# Use of Cells, Supplements, and Peptides as Therapeutic Strategies for Modulating Inflammation after Spinal Cord Injury: An Update

**DOI:** 10.3390/ijms241813946

**Published:** 2023-09-11

**Authors:** Elisa Garcia, Vinnitsa Buzoianu-Anguiano, Raúl Silva-Garcia, Felipe Esparza-Salazar, Alejandro Arriero-Cabañero, Adela Escandon, Ernesto Doncel-Pérez, Antonio Ibarra

**Affiliations:** 1Centro de Investigación en Ciencias de la Salud (CICSA), FCS, Universidad Anáhuac México Campus Norte, Huixquilucan 52786, Mexico; edna.garcia@anahuac.mx (E.G.); felipe.esparzas69@anahuac.mx (F.E.-S.); adela.escandon.cesarman@gmail.com (A.E.); 2Grupo Regeneración Neural, Hospital Nacional de Parapléjicos, SESCAM, 45071 Toledo, Spain; vbuzoianu@externas.sescam.jccm.es (V.B.-A.); aarrieroc@externas.sescam.jccm.es (A.A.-C.); 3Unidad de Investigación Médica en Inmunología Hospital de Pediatría, CMN-SXXI, IMSS, Mexico City 06720, Mexico; silgarrul@yahoo.com.mx

**Keywords:** spinal cord injury, cell therapy, neuroprotection, supplements, peptides, inflammation

## Abstract

Spinal cord injury is a traumatic lesion that causes a catastrophic condition in patients, resulting in neuronal deficit and loss of motor and sensory function. That loss is caused by secondary injury events following mechanical damage, which results in cell death. One of the most important events is inflammation, which activates molecules like proinflammatory cytokines (IL-1β, IFN-γ, and TNF-α) that provoke a toxic environment, inhibiting axonal growth and exacerbating CNS damage. As there is no effective treatment, one of the developed therapies is neuroprotection of the tissue to preserve healthy tissue. Among the strategies that have been developed are the use of cell therapy, the use of peptides, and molecules or supplements that have been shown to favor an anti-inflammatory environment that helps to preserve tissue and cells at the site of injury, thus favoring axonal growth and improved locomotor function. In this review, we will explain some of these strategies used in different animal models of spinal cord injury, their activity as modulators of the immune system, and the benefits they have shown.

## 1. Introduction

Spinal cord injury (SCI) can occur at any level or segment [1]. The pathological events originated after injury cause two types of damage. The first refers to direct mechanical injury of the spinal cord that anatomically causes contusion, compression, hemisection, or complete transection of the spinal cord. These inductors of damage have similar self-destructive mechanisms, but their progression and complications are different [2]. In general, the primary injury directly imparts force to the spinal cord, disrupting axons, blood vessels, and cell membranes. This primary damage eventually triggers a cascade of harmful events like vascular dysfunction, edema, ischemia, excitotoxicity, free-radical production, inflammation, and delayed apoptotic cell death that, over a period of time, extend tissue damage causing the secondary injury. Whereas neurological deficits are present immediately following the initial injury, the secondary injury results in a protracted period of tissue destruction.

On the other hand, based on the time after injury and pathologic mechanisms, this injury process can be divided into acute, subacute (or intermediate), and chronic phases. In each one, there are diverse destructive mechanisms with a different degree of damage that determine neuronal destruction and allow the possibility of finding therapeutic targets to promote neuroprotection or neuroregeneration [3]. One of the most deleterious phenomena after injury is inflammation. The early acute phase involves the infiltration of inflammatory cells and continuous activation of resident microglia. The inflammatory process following SCI is highly complex and involves numerous cellular populations, including astrocytes, microglia, T cells, neutrophils, and peripheral monocytes, that are known to participate in the major inflammatory reactions. In this acute phase of injury, various inflammatory events contribute to neuronal and glial destruction. These phenomena are initiated by the primary insult, which leads to the release of significant amounts of DAMPs (damage-associated molecular patterns) and continues into the secondary phase of damage. This inflammatory reaction involves the activation of both resident and peripheral cells, such as systemic macrophages and neutrophils. These cells access the site of the lesion through locally produced chemokines and ICAM (intercellular adhesion molecules). Additionally, resident cells, such as microglia and astrocytes, play a role in the acute and subacute phases, through their activation toward inflammatory phenotypes, which are characterized by the production of proinflammatory molecules (proteases, myeloperoxidase, reactive oxygen (ROS) and nitrogen (RNS) species, inducible nitric oxide synthase (iNOS), nitric oxide (NO), macrophage inflammatory protein 1 (MIP 1 α, γ, β), monocyte pro-chemoattractant-1 (MCP-1), C-X-C motif chemokine ligand-10 (CXCL-10), IL1α/β, tumor necrosis factor-alpha (TNFα), and IL6) and the presentation—by major histocompatibility complex II (MHC II)—of antigens derived from the central nervous system (CNS) like myelin basic protein (MBP). This presentation phase exacerbates the inflammatory response by promoting the activation of helper T cells to the Th1 phenotype, which release additional proinflammatory cytokines (TNFβ, interferon-gamma IFNγ, and IL12) and B cells, leading to the production of IgG and IgM antibodies. Both T cells and B cells will favor the increase in the inflammatory response through the release of inflammatory cytokines, which further differentiate microglia and macrophage cells into M1 inflammatory phenotypes. These cells promote lipid peroxidation due to excessive NO and superoxide anion production, resulting in the generation of a neurotoxic compound called peroxynitrite, which is responsible, in part, for Wallerian degeneration [4,5,6].

On the other hand, chemokines direct immune cells to the site of injury and activate a signaling cascade that increases the permeability of neuronal cell membranes and the release of glutamate. Overactivation by glutamate, also known as “excitotoxicity”, results in increased calcium influx, triggering free-radical production and cell death by pro-apoptotic reactions. Cell death is not limited to damaged neurons but can also affect surrounding myelin-producing oligodendrocytes and healthy neurons, affecting up to four segments of the trauma site [7].

The physiological response after injury intends to resolve the inflammatory response through the release of anti-inflammatory cytokines, such as IL10 and TGF beta, by reactive astrocytes. Nevertheless, far from solving the problem, these cytokines stimulate the proliferation of astrocytes, leading to the formation of a glial scar. Additionally, these cytokines could contribute to fibroblast proliferation and collagen production. During the chronic phase, the glial scar indirectly forms a barrier at the lesion site, preventing neuronal reconnection and axonal conduction. Under these conditions, other molecules like growth factors (brain-derived growth factor (BDNF), neural growth factor (NGF), epidermal growth factor (EGF), and insulin growth factor (IGF)) and neurotrophins (NT3, 4, and 5), are produced with the aim of promoting neural restoration; they are mainly produced by the M2 phenotype of macrophages and microglia and Th2 lymphocytes. Unfortunately, these cells are not stimulated at the optimal time and quantity to promote their beneficial effects. In contrast, in the center of the injury, the M1 and Th1 phenotypes are predominantly activated, causing more tissue damage [4,5,6] (Figure 1).

Regrettably, there are currently no established methods for reversing spinal cord damage. Nevertheless, scientists are persistently engaged in developing novel therapies, such as prosthetics and drugs, aimed at encouraging the regeneration of nerve cells or enhancing the performance of nerve cell projections (axons) after SCI. Presently, the treatment of SCI is centered around averting additional harm and enabling individuals with an SCI to reintegrate into vibrant and meaningful lives [3].

Upon the patient’s hospital admission, medical professionals focus on sustaining the individual’s ability to breathe and immobilizing the neck to prevent further damage to the spinal cord. Moreover, healthcare providers address acute injuries through methods such as surgery, which may involve decompression laminectomy to relieve pressure on the spinal cord, the removal of bone fragments or foreign objects, spinal fusion, or the application of a brace. Traction is also utilized to stabilize the spine and restore alignment. Administering Methylprednisolone within 8 h of the injury has been shown to improve certain patients by reducing nerve cell damage and inflammation around the injury site. Additionally, ongoing research is exploring experimental treatments to prevent cell death, manage inflammation, and promote nerve repair or regeneration. However, these studies must receive approval at the clinical level and be offered to patients within the institution where they are being treated if the patient is interested in participating. Several of these strategies are progressing toward clinical studies, including trials that have received approval from the US Food and Drug Administration (FDA). These trials are poised to ultimately ascertain the safety and effectiveness of these approaches [11,12,13].

Individuals with an SCI can derive benefits from rehabilitation, encompassing physical therapy to enhance muscle strength, communication, and mobility, as well as incorporating assistive tools like wheelchairs, walkers, and leg braces. Adaptive devices aid in communication, while occupational therapy refines fine motor skills and self-care techniques. Coping mechanisms for spasticity and pain, vocational therapy with assistive devices for returning to work, and recreational activities are also a part of rehabilitation. Additionally, strategies for exercise and healthy diets are emphasized owing to obesity and diabetes posing potential risks. Functional electrical stimulation has potential for restoring neuromuscular, sensory, and autonomic functions, such as bladder, bowel, and respiratory functions [12,13].

Various pharmacological strategies have been developed to reduce, modulate, or inactivate inflammation after injury; however, the results have not been very satisfactory in inducing a better functional recovery. For these reasons, other therapeutic alternatives continue to be sought. In line with this, non-pharmacological strategies have also been a promising alternative to modulate the inflammatory response in the different phases of SCI (see Figure 1). Some of the most studied strategies include cell therapy and the use of immunomodulatory peptides and supplements (including vitamins, minerals, and probiotics). In this review we analyzed the most important of them.

## 2. Cells Useful in Therapy for Spinal Cord Injury

Cell transplantation has been used to repair the injured spinal cord both in the acute and chronic phases, and the use of allogeneic transplantation, including different cell types, is of great interest as it is a positive alternative for axonal regeneration. In line with this, Schwann cells (SCs), olfactory glia ensheathing cells (OECs), hematopoietic embryonic stem cells, neural stem cells, and bone marrow stromal cells [14] have been used. The importance of this therapy is the different mechanisms that favor an improvement in function after damage, including neuroprotection, immunomodulation, axonal branching, axonal regeneration, and axonal remyelination [15].

### 2.1. Schwann Cells

Schwann cells (SCs) are a part of the peripheral nervous system (PNS), which is responsible for forming myelin sheaths on peripheral axons [16]; unlike oligodendrocytes in the CNS, SCs have a role in the repair of nervous tissue. After an injury, they can change their phenotype from myelinating to phagocytic and recruit neutrophils and macrophages through chemotactic signaling, which is the key to the regenerative capacity of PNS [17]. This is why SCs have a strong relationship with the immune system [18] because they are modulated through the exchange of different signals.

In PNS injuries, SCs cells have been observed to be able to modulate macrophages and mediate their transition from M1 to M2, although not through classical M2-mediated cytokines [19], as during the peak of debris clearance, this macrophage phenotype is increasing owing to elevated arginine expression [20].

SCs are capable of both recognizing and presenting antigens, as well as expressing many cytokines, chemokines, and immune modulatory factors and responding to them through the expression of their receptors [18,21]. Toll-Like Receptors (TLRs) enable the expression of TNF-α, iNOS, and MCP1, which promotes macrophage recruitment. Pattern Recognition Receptors (PRRs) have been shown to induce the immune response by modulating T lymphocytes through the presentation of antigens in the context of MHC II. On the other hand, factors, such as NGF, that are secreted by SCs through the p75/AMPK/mTOR receptor promote increased autophagy of myelin debris [22].

The transplantation of SCs can depend on the stage of the spinal injury and achieve immunomodulation and neuroprotection of the resident tissue [23,24]. In animal models with a spinal contusion, SC transplantation has been shown to reduce the number of cystic cavities and increase the level of neuronal marker NeuN+ [25]. In the subacute phase, SCs also decrease the number of cells expressing CD11b+, CD68+, and Iba1+ cells, reducing the number of proinflammatory macrophages/microglia [26]. The Mousavi’s group tested the neuroprotective potential of SCs by combating inflammasome activation, demonstrating improved motor function and remyelination, because SCs reduced levels of NOD-like receptor Pyrin domain-containing 1a (NLRP1a) and NLRP3 (inflammasome activators), caspase 1 (Casp1), TNF-α, and IL-1β [27].

SCs play a role in remyelination and neurogenesis during SCI repair. Isolated SCs retain the functional capacity for nerve repair and remyelination [28]. After SCs are transplanted into the injured CNS, they can reduce the size of lesions, attract and guide descending and ascending axons to the implant, and improve locomotor function without additional interventions [28]. In addition, it has been shown that the co-transplantation of SCs with neural precursor cells, derived from human embryonic stem cells, could offer a synergistic effect and promote neuronal differentiation and functional recovery [29].

### 2.2. Olfactory Ensheathing Cells

OECs are glial cells that envelop bundles of olfactory axons, both peripherally in the olfactory nerve and within the olfactory nerve layer (ONL) of the olfactory bulb [30]. OECs play a critical role in the growth of neurites and establishment of functional connections along the olfactory plaque, where new olfactory sensory neurons are generated from olfactory epithelial stem cells [31]. These cells can respond to invasion or damage and modulate the immune system, releasing IL-10 and TGF-β, promoting phagocytosis via integrin receptors [32], and having receptors that help mediate the release of cytokines, such as IL-6, ciliary neurotrophic factor (CNTF), and leukemia inhibitory factor (LIF) [32,33].

Several studies are using OECs as a treatment for SCI, in different animal models, not only because they can modulate the inflammation of the injury but also owing to their regenerative capacity [34]. In transection models of SCI, the transplantation of OECs decreases immune cell infiltration by interacting with astrocytes via the signal transducer and activator of transcription 3 (STAT3) pathway, reducing the cystic cavity, and protecting and preserving axons [35]. They can also induce the polarization of microglia from M1 to M2 by inhibiting the JAK/STAT3 pathway, and they reduce the population of the proinflammatory iNOS+/CD86+ phenotype and increase the anti-inflammatory microglia/macrophage phenotype Arginase+/CD206+ [32]. Moreover, they can reduce TNF-α, IL-1β, and oxidative stress levels and elevate the IL-10 level to protect nerve tissue [36]. In addition, OECs can phagocytose myelin debris, increasing neuronal survival via the p38/MAPK pathway [37], and even the transplantation of their exosomes alone promotes the M1/M2 switch of microglia/macrophages, inhibiting the nuclear factor kappa beta (NFkB) pathway [38]. The Lopez-Vales group observed that transplanted OECs modulated the early astrocyte response from A1 to A2 astroglial phenotype. The transplanted OECs reduced the inflammatory peak to stop the earlier inflammatory phase and glial scarring, thus slowing down degeneration and preventing the spread of spinal cord damage [39].

In the subacute phase, studies have shown that they can reduce the number of astrocytes and macrophages by decreasing infiltration; reducing CCL2/3 chemokine levels; modifying cytokine levels by increasing levels of anti-inflammatory cytokines, such as IL-10/13; and decreasing inflammatory IL-6 and TNF-α levels. Additionally, OECs are capable of modifying the phenotype of astrocytes and macrophages, down-regulating the number of iNOS+/CD16+/32+ macrophages, and increasing Arginina+/CD206+ levels through the secretion of IFN-γ and IL-4 [40].

### 2.3. NG2 Glia or Oligodendrocyte Precursor Cells

NG2 glia, also known as oligodendrocyte precursor cells (OPCs), are located throughout the CNS and serve as a reservoir of precursors to differentiate into oligodendrocytes. In response to SCI, NG2 cells increase their proliferation and differentiation into myelinating oligodendrocytes [41]. However, post-traumatic endogenous remyelination is rarely complete, and a better understanding of the characteristics of OPCs and their manipulation is critical for the development of new therapies [42,43]. The role of these cells in neuroprotection and modulation of the immune system in injuries has been studied, but only their endogenous effect.

In IL-1β^−/−^ knockout mice, a demyelination process arose, and OPCs were not capable to differentiate and maturate into myelinating oligodendrocytes, which means that IL-1β is required for OPC differentiation. In addition, in contusion models, C-X-C chemokine receptor type 4 (CXCR4) and C-X-C Motif Chemokine Ligand 12 (CXCL12) expressions are upregulated in astrocyte-like cells [44]. Although the activation of CXCR4 by CXCL12 promotes the differentiation of OPCs under demyelinating conditions [45], despite this result, the functional significance of these changes following injury is still unknown. Finally, IL-17 increases the production of IL- 6 and matrix metalloproteases (MMP3 and MMP9) by OPCs in vitro, suggesting that OPCs may be directly involved in regulating the inflammatory response to injury [46].

The reduction in the accumulation of activated microglia and macrophages at the site of injury decreases the proliferative response of OPCs to the site of injury by eliminating β-catenin, specifically in OPCs [47], raising the possibility that leukocytes and monocytes can be attracted to the site of injury. OPCs were also found to be important for the maintenance of microglial homeostasis because in their absence, microglia were driven toward a reactive phenotype [48]. In addition, they regulate innate immunity as they can secrete TGF-2, which inhibits microglial activation via the SMAD2 (mothers against DPP homolog 2) pathway [40].

### 2.4. Bone-Marrow-Derived Mesenchymal Stem Cells

Bone-marrow-derived mesenchymal stem cells (BMSCs) are adult multipotent cells that have the capacity for self-renewal, proliferation, and differentiation. BMSCs are an alternative for the experimental treatment of SCI (contusions, complete sections, or ischemia). They have been shown to promote axonal regeneration and improvement in locomotor function [49,50,51,52]. They also can form bundles of cells that serve as a bridge at the epicenter of the SCI [49,53,54]. In consonance with the context, they can differentiate both in vitro and in vivo into cells that express markers related to astrocytes, oligodendrocytes, SCs, microglia, and neurons [55,56]. In their niche, they integrate endocrine, autocrine, and paracrine signals by secreting factors that control hematopoietic cells, keeping them in an undifferentiated state by secreting stem cell trophic factor (SCF), stromal cell factor (SDF-1 or CXCL12), bone morphogenetic protein 4 (BMP-4), LIF, and granulocyte monocyte colony-stimulating factor (GM-CSF) [57].

In contusion and compression models of SCI, the use of BMSCs favors the decrease in proinflammatory cytokines, such as TNF-α and IL-6, and promotes the secretion of anti-inflammatory cytokines, such as IL-4, IL-10, and IL-13, which favors the activation of M2 macrophages, supporting a neuroprotective environment [58,59]. They can increase the number of M2 macrophages and decrease the number of M1 macrophages at the site of injury, which may contribute to improved function by protecting healthy axons at the site of injury [16].

BMSCs have been shown to inhibit T-cell activation and division by down-regulating cyclin D2, promoting the expression of p27Kip1 in the G1 phase of the cell cycle. The inhibitory effect on T cells is mediated by the secretion of TGF-β, hepatocyte growth factor, and prostaglandin E2. In addition, BMSCs secrete soluble factors as inflammatory mediators, such as indoleamine 2–3 dioxygenase (IDO), iNOS, and homo-oxygenase 1; they can also secrete human leukocyte antigen G and are involved in the inhibition of CD4+ and CD8+ T cells by inhibiting STAT5 phosphorylation [16,57,60]. They can not only influence the switch from Th1 to Th2 but also decrease the Th17/Treg ratio, accompanied by a decrease in TNF-α and IL-1β, which increases TGF-β levels [16]. Finally, BMSCs have been reported to have a neuroprotective capacity following injury as a positive strategy for CNS repair.

### 2.5. Neural Stem Cells

Neural stem cells (NSCs) are precursor cells located in the lateral ventricle of the brain, dentate gyrus of the hippocampus, and central canal of the spinal cord [61]. They are capable of self-renewal and differentiation into oligodendrocytes, astrocytes, and neurons [62]. The main mechanism of these cells in neurodegenerative diseases is the modulation of astrocytes on glial scar formation, promoting differentiation to oligodendrocytes and neuronal differentiation, nerve cell replacement after SCI, and secretion of pro-regenerative factors to protect damaged tissue by promoting neuritic growth [61].

NSCs can secrete multiple growth factors, such as BDNF, CNTF, glial-derived neurotrophic factor (GDNF), NGF, and IGF-1, which contribute to the survival and growth of neuronal cells. After injury, they regulate T cells and macrophages to inhibit demyelination by reducing the number of CD4+ T cells, favoring a regenerative phenotype shift [61,62].

In addition, they have been shown to suppress the accumulation of neutrophils and macrophages at the site of injury, causing M1 macrophage activation to be blocked and a decrease in the secretion of proinflammatory cytokines, such as TNF-α, IL-6, IL-1β, and iNOS. This change in M1 macrophage polarization is due to the secretion of factors, such as type 1 metalloproteinase (TIMP-1), vascular endothelial growth factor (VEGF), TGF-β, MMP9, and haptoglobin [62]. They have also been found to affect neuronal apoptosis by reducing glutamate exposure, decreasing levels of pro-apoptotic markers, and increasing levels of anti-apoptotic markers, such as B-cell lymphoma 2 (Bcl-2) [63].

### 2.6. Neural Precursor Cells: Aldynoglia

In the spinal cord and brain, we can find aldynoglia cells that make up the ependymal layer of the spinal cord. Within these, tanycytes, ependymal cells, and central canal neurons could be presented in an ependymal layer of the spinal cord, called the spinal ependymal layer (SEL) [64].

In the subacute phase, the SEL migrates to the site of injury to begin proliferation and differentiation and may also interact with immune response cells. The polarization of M1/M2 macrophages/microglia at the site of injury has been shown to directly affect the growth and differentiation of ependymal stem cells (EpSCs), specifically, M2 polarization, promoting neuronal differentiation [65].

The mechanism by which M2 regulates the differentiation of EpSCs is through (Sirtuin-2) SIRT2. SIRTs, NAD+-dependent diacetylated histone class III, are involved in catalyzing several biological processes, such as metabolism and gene expression [64,65]. In vivo and in vitro studies have shown that M2 microglia promote SIRT2 upregulation of EpSCs, directly affecting their differentiation; this change acts directly on microtubule dynamism, thus, promoting their differentiation into neurons. This change is produced by growth factors, such as BDNF, which is secreted by M2 microglia, activating the BDNF/tropomyosin receptor kinase B (TrkB)-signaling cascade when the TrkB tyrosinase B on the surface of EpSCs is activated and inducing the expression of SIRT2 [64,65].

The combination of immune responses and effects after SCI of all the mentioned cell types are summarized in Table 1.

As we have already shown, the use of cells in animal models of spinal cord injury favors neuroprotection and, therefore, promotes better functional recovery. Demonstrating this, clinical studies with patients have shown that transplants have not generated adverse events that are a clinical risk for patients. In this context, SCs isolated from sural nerves were used for autotransplantation, minimizing risks and showing the safety and viability of these cells and functional improvement in some cases [66]. After two years of treatment, some patients improved their motor, sensory, and sphincter functions. Gant’s group highlighted a case that improved one point at the neurological level, six points in sensory function, and four points in motor function [67].

Other cell types are OECs that have been used in studies of patients with a chronic spinal cord injury; the use of these cells has not shown risks or secondary consequences after transplantation. In addition, different authors have described that patients achieved an improvement in motor function, as assessed with the ASIA, in the recovery of sensation and bladder function [68,69].

OPCs have been used by Fessler’s group in a phase 1/2a study, using three different doses of OPC (LCTOPC1, known as GFNOPC1 or AST-OPC1) administered during the subacute phase in cervical lesions. The authors demonstrated that after a follow-up year, clinical administration was safe. Thirty-two percent of patients recovered two or more neurological function points on at least one side of the body, and 96% improved one or more neurological function points [70].

The most widely used type of cells is mesenchymal stem cells as a treatment for patients with spinal cord injury because these cells are easy to obtain from different tissues and have not presented risks for patients because most are autotransplants [71]. It has been shown that they favor the improvement of motor and sensory functions. We will show some of the studies of each type of cell used in patients in Table 2.

The neuroprotective and neurorestorative effects shown by cell therapy after SCI are encouraging; however, each type of cell needs a specific microenvironment to exert its beneficial effects or induce changes in immune cell phenotypes, as previously mentioned in this Section. However, to enhance this attribute, the use of immunomodulatory peptides, supplements, probiotics, or prebiotics could also be considered in a combined therapy.

## 3. Immunomodulatory Peptides after Spinal Cord Injury

The use of peptides as a therapeutic strategy after SCI has shown encouraging results. The findings have encouraged the search for a different innovative therapy where the inflammatory response and promotion of neuroregeneration are the main variables. Peptides currently are demonstrating high activity, safety, low cost, and easy production [87]. An increasing number of active peptides have been studied, including antioxidants, analgesics, immunomodulators, and others. In the context of this review, our focus is directed to certain peptides unrelated to myelin or associated with myelin limited to animal studies. This approach is taken as clinical studies have not undergone approval.

### 3.1. Non-Myelin-Related Peptides

#### 3.1.1. Glutathione Monoethyl Ester (GME)

Glutathione is a tripeptide formed by the amino acids glycine, cysteine, and glutamic acid. It is produced by the liver and involved in many bodily processes. Glutathione functions as a cell reducer, as a catalyst in various biological and metabolic reactions, and in the protection against ROS and toxic compounds of endogenous and exogenous origins [88]. The possibility that an increase in cellular glutathione levels may be beneficial under certain conditions to protect cells against oxidative stress means that glutathione is a potent antioxidant; however, its depletion is significant when used, and its recovery and transport to cells is limited. Glutathione monoethyl ester (L-γ-glutamyl-L-cysteinyl glycine ethyl ester) is efficiently transported to many cells, and glycine carboxylic ester is deesterified intracellularly in cells and rapidly converted to glutathione, using approximately 90% of the glutathione monoethyl ester administered, and subjected to deesterification after 30 min to obtain a high bioavailability as an antioxidant [88]. Glutathione, in its reduced form (GSH), is quantitatively the most important endogenous rechargeable antioxidant. It has also been shown that it acts as a vasodilator under conditions of oxidative stress that alter endothelial function, thus improving cell ischemia [89]. Furthermore, it performs numerous crucial roles connected to transporting amino acids across membranes, synthesizing and breaking down proteins, regulating genes, and managing cellular redox processes [88] by the oxidative stress that occurs after CNS trauma [90]. Reduced glutathione monoethyl ester (GSHE) is permeable to cells and efficiently transported to the cerebrospinal fluid [88], thus providing the most direct and convenient means available for increasing the GSH concentration in an intracellular manner [91]. Some studies have evaluated the significant effect of GSHE on lipid peroxidation (LPO) after SCI [92] and on neuroprotection after transient focal cerebral ischemia. GSHE is capable of reducing infarction sizes from 46% to 16% [88].

Following spinal cord injury, the utilization of GSHE was contrasted with methylprednisolone (MP). MP is presently the sole approved immediate treatment accessible for individuals with SCI. The use of corticosteroids in the treatment of SCI has been widely studied. It is currently known that their administration could be beneficial owing to the anti-inflammatory properties of these drugs, reducing the formation of edema generated, in turn, by the local inflammatory process [93,94]. MP exerts a neuroprotective effect by suppressing the tissue inflammatory response through the inhibition of the inflammatory cell function, including chemotaxis, phagocytosis, synthesis of inflammatory mediators, and lysosomal enzyme release [95,96,97]. Several clinical studies on the use of MP question the effectiveness of this treatment and the safety of long-term side events in patients; however, it is currently still used in common practice, and, for legal medical reasons, it is the appropriate treatment [93]. To determine if GSHE has more beneficial effects than MP, GSHE was administered intraperitoneally to rats subjected to moderate SCI. The animals receiving GSHE showed significant recovery of motor function, increased postoperative body weight, and survival of red nucleus neurons compared to the MP and control groups [98].

#### 3.1.2. Monocyte Locomotion Inhibitory Factor

Another peptide used like a novel strategy is monocyte locomotion inhibitory factor (MLIF), which is an exogenous pentapeptide (Met-Gln-Cys-Asn-Ser) produced by Entamoeba histolytica and has a potent anti-inflammatory effect (Silva-Garcia and Rico-Rosillo, 2011). In vitro studies in peripheral blood cells (monocytes and neutrophils), cell lines of human promonocytes (U937), and fibroblasts (MRC-5) treated with MLIF demonstrated that the factor decreases or inhibits: (1) the expression of the chemokines CCL3, CCL4, CCL1, and the CCR1 receptor, which are mainly involved in the chemotaxis of monocytes; (2) the synthesis of ROS and RNS; and (3) the production of IL-1β, IL-6, IL-12, and IFN-γ [99,100]. On the other hand, MLIF favors the synthesis of IL-10 and TGF-β and translocation to the nucleus of the p50/p50 homodimer of NF-κB, suggesting that this translocation reduces inflammatory gene expression [100].

The neuroprotective effect of MLIF has been demonstrated in the C57BL/6 mouse model infected with *Plasmodium berghei*. In this case, the use of this peptide prevented increases in TNF-α and IFN-γ, protecting the integrity of the blood–brain barrier and providing an increase in the survival of animals without showing signs of brain damage [101].

Lipid peroxidation is a neurodegenerative phenomenon that occurs after SCI, caused by increased production of ROS an RNS, such as NO. Recently, in an experimental model of traumatic SCI with the application of intraspinal MLIF, the peptide caused a downregulation in the expression of the iNOS gene that was correlated with a lower systemic production of NO and lipid peroxidation. MLIF also increased the expression of neuroprotective anti-inflammatory cytokines, such as IL-10 and TGF-*β*; these findings correlated with the preservation of neurons of the ventral horn and rubrospinal tract and a significant improvement in neurological recovery [102].

### 3.2. Myelin-Related Peptides

Several neuroprotective or neurorestorative strategies focus on preserving myelinated axons or inducing axonal growth to promote axonal reconnection and then motor and functional recovery. MBP is produced by oligodendrocytes and by SC. This protein is localized in the myelin sheath, which is a unique multilayered membrane that surrounds the axon of neurons in which the cytoplasmic and extracellular regions accumulate alternately and compactly [103]. This unique myelin structure provides several neurological functions, including saltatory conduction, nerve metabolism, ion regulation, and water homeostasis [104,105]. After SCI, the processes of lipid peroxidation and T-cell recognition of the MBP lead to axon demyelination. This canonical recognition of self-constituents allows the participation of an autoreactive response in the neurodegenerative process observed after injury. The activation of T cells against neural antigens—especially against MBP—intensifies the inflammatory response and, thereby, the destruction of neural tissue. That is why several therapeutic strategies were directed to inhibit these autoreactive and inflammatory responses. Nevertheless, as immune cells are necessary for protecting and remodeling tissues, the elimination of the immune function at the site of injury was not the best strategy [106]. It was more realistic to look for the modulation instead of the elimination of the immune function. For this reason, different neural-derived peptides were generated to modulate the autoreactive response [107,108,109,110].

Presently, it is widely acknowledged that a well-regulated inflammatory response subsequent to the injury is indispensable for safeguarding neural tissue and promoting recovery. In the aftermath of spinal cord injury (SCI), an extended and unregulated reaction unfolds, initiating a detrimental cycle of heightened glial sensitivity that accentuates harm to neurons. In 1999, the research team of Dr. Michal Schwartz disclosed that autoimmune reactions within the central nervous system (CNS) might, under specific circumstances, shield injured neurons against the propagation of harm and expedite the regenerative mechanisms within the wounded spinal cord [85,86]. They introduced the novel notion of protective autoimmunity (PA) to denote the “autoimmune reaction that triggers CNS defense against injury when local non-immunological protective mechanisms falter in sufficiently counteracting injury-induced toxicity” [85]. Subsequent investigations revealed that immunization using myelin-derived proteins or altered neural-derived peptides could yield significant advancements in enhancing motor functionality following SCI [87,88].

Inducing an appropriate protective autoimmunity (PA) through immunization using a milder variant of the autoantigen can effectively regulate inflammation and yield favorable effects. In recent times, extensive research has been conducted to formulate these immunomodulatory peptides and explore their potential therapeutic uses. Glatiramer acetate (GA) and A91 have been employed to trigger PA in animal models subsequent to the SCI, demonstrating promising outcomes either individually or when combined with other therapeutic approaches.

#### 3.2.1. Glatiramer Acetate

GA, also called copolymer 1 (Cop-1), is a random copolymer comprising glutamic acid, lysine, alanine, and tyrosine with an average molar fraction of 0.141, 0.427, 0.095, and 0.0338, respectively. GA has been approved by the Food and Drug Administration (FDA) for the treatment of relapsing–remitting multiple sclerosis [111,112]. Although its exact mechanism of action remains partially understood, it appears that GA possesses immunomodulatory characteristics along with neuroprotective attributes [91]. GA has demonstrated the ability to elicit an immunomodulatory response within both innate and adaptive immune cell populations through the inhibition of the activation of T cells reactive to MBP (myelin basic protein) and the promotion of an anti-inflammatory environment within T cells. Repeated immunization with glatiramer acetate may modulate the adaptive immune system by shifting from a proinflammatory Th1 immune response to an anti-inflammatory Th2 phenotype characterized by IL-4 secretion [112].

GA has been assumed to be an immunomodulatory agent with neuroprotective properties in various neurodegenerative diseases, including Alzheimer’s disease, Parkinson’s disease, and amyotrophic lateral sclerosis [113,114,115,116]. Furthermore, it has been established that GA also wields a suppressive impact on the M1 phenotype of microglia while concurrently encouraging the M2 phenotype. GA’s capacity to safeguard neurons and oligodendrocytes is well-documented, influencing three distinctive aspects of neurogenesis: the proliferation of neuronal precursor cells and their migration and subsequent differentiation [91,96,97,98,99].

The efficacy of GA in the context of spinal cord injury (SCI) appears to hinge on factors, such as the dosage administered, timing of treatment initiation, and severity of the lesion. An investigation into the impact of GA treatment utilizing lower doses over a span of two weeks (0.5 mg/animal/day) was conducted in rats subjected to ventral lumbar root avulsion. The outcomes revealed that GA treatment led to a notable 40% enhancement in neuronal viability within the motor nucleus of the spinal cord, coupled with a decrease in astroglial reactivity at the injury site [100].

On the other hand, immunization with GA showed that it reduces lipid peroxidation. It is possible that GA is interfering with NO production. To test this hypothesis, the effect of GA on the amounts of NO and iNOS expression were evaluated on glial cells when co-cultured with autoreactive T cells and cells from spinal cords of injured animals (mice and rats). In vitro studies showed that GA significantly reduced the production of NO by glial cells. This observation was substantiated by in vivo experiments demonstrating that immunization with GA decreases the amounts of NO and iNOS gene expression at the site of injury; these results clarified a possible mechanism by which protective autoimmunity promotes neuroprotection [117].

In a study more recently conducted, the administration of GA treatment was carried out in female Sprague–Dawley rats. A substantial dose of GA (2 mg/kg) was administered continuously for 28 days following the occurrence of injury. Notably, different from prior research, the authors observed a hindrance in locomotor recovery, a heightened occurrence of neuronal loss during the acute phase following the spinal cord injury (SCI), and an adverse reaction targeting MBP [102].

The controversial results observed in the former studies could be caused by diverse factors. For instance, the intensity of injury plays a crucial role in the outcome of injury. In line with this, a study on lesions of different intensities (moderate and severe) showed changes in the expression of eight different genes: IL6, IL12, IL1β, IFNγ, TNFα, IL10, IL4, and IGF-1. Sprague–Dawley females were immunized with Cop-1 (150 μg/animal) after injury, and changes in gene expression were obtained at 7 days [118]. It was shown that a moderate lesion allows Cop-1 to create a microenvironment where cytokines, such as IL4 and IL10, prevail, which could play an important role in the protection and restoration of neural tissue [119,120]. Aside from this, a significant reduction in inflammatory cytokines, such as TNFα, was observed [121,122]. In contrast, after a severe contusion, Cop-1 failed to induce the same effect.

In this context, compared with moderate SCI, severe injury causes a more pronounced release of DAMPs and neural components so that the high concentration of these molecules could be modifying the immune response to a Th1 encephalitogenic phenotype [123]. This predominant phenotype, which could be directed against other immunogenic determinants, could also inhibit the proliferation of protective Th2 lymphocytes elicited by Cop-1 and, therefore, their beneficial actions [117].

#### 3.2.2. A91 Peptide

A91, a peptide derived from the immunogenic sequence [87,99] of MBP, features a substitution of alanine for the lysine residue at position 91. Upon inoculation, A91 prompts the proliferation of CNS antigen-specific T cells that enact protective measures through diverse mechanisms fostering neuroprotection [83,109].

In an endeavor to establish prophylactic therapy involving A91, a meticulously designed study encompassed rat subjects immunized with the peptide before the onset of SCI. This immunization resulted in a significant rise in the survival rate of rubrospinal and ventral horn neurons, with a corresponding substantial enhancement in motor recovery [88].

Several mechanisms underlying the A91 immunization’s neuroprotective and neurorestorative properties have been documented. Once presented to CD4+ helper T lymphocytes, A91 orchestrates the moderation of inflammation in the subacute phase of SCI by inciting Th2 lymphocytes—an anti-inflammatory phenotype [101]. Cytokines produced by A91-activated lymphocytes, primarily IL-4 and IL-10, work to reduce ROS levels. Elevated IL-10 levels not only curtail TNF-α synthesis in macrophages but also partake in the inhibition of lipid peroxidation. Conversely, IL-4 diminishes the production of IFN-γ, a cytokine that propels macrophages and microglia toward a proinflammatory M1 phenotype. Additionally, IL-4 encourages arginase production, which, in turn, lowers NO production by iNOS via arginine removal. Consequently, the presence of fewer nitrates in the lesion area curbs RNS formation [8,110].

A91 immunization has also been found to curtail apoptosis resulting from traumatic injury. This reduction is paralleled by a noteworthy decrease in caspase-3 (Casp3) activity and TNF-α concentrations [111]. Furthermore, A91 immunization triggers the generation of neurotrophic factors vital for optimal CNS function. These factors contribute not only to tissue protection, regeneration, and neurogenesis but also to the initiation of new synaptic connections [112,113]. A91 prompts significant production of anti-inflammatory proteins linked to regeneration and a notable increase in neurogenesis during the chronic stages of the lesion [114]. Additionally, it facilitates a lasting improvement in the production of BDNF, NT3, and GAP-43. This favorable outcome is tied to enhanced motor and sensory recoveries during the chronic phase of SCI [112].

The therapeutic strategy of A91 immunization has produced promising outcomes in both acute and chronic SCI scenarios, the latter of which is challenging to treat successfully. However, these beneficial effects have not been observed in animals with severe SCI [103,112]. Previous research highlights that severe injury or excessive A91 administration hampers the positive effects of protective autoimmunity [101,115]. Moreover, animals subjected to severe SCI after A91 immunization showed negligible IL-10 production, suggesting that the beneficial effects are hindered and that an inflammatory microenvironment dominates in severe SCI [103].

Although A91 immunization is promising, its effectiveness could be enhanced by combining it with other therapeutic strategies.

##### Synergistic Effects of A91 Immunization with Other Strategies

Given the encouraging outcomes of A91 immunization, the potential benefit of this therapy in combination with methylprednisolone (MP) treatment was explored. Investigations indicated that when MP was administered immediately after SCI and vaccination with A91 followed 48 h later, the beneficial effect of A91 immunization was not compromised by MP [74].

In another study, A91 immunization was coupled with GSHE, an antioxidant known for its neuroprotective effects, following SCI. This combined approach demonstrated that after SCI, the combination therapy outperformed the standalone immunization, leading to improved motor recovery, a higher count of myelinated axons, and enhanced survival of rubrospinal neurons. GSHE’s action in reducing ROS levels and lipid peroxidation does not interfere with the therapeutic effect of A91, thereby establishing the superiority of the A91-GSHE combination when administered shortly after SCI or within the initial 72 h post-injury period [115,116].

Recently, it was shown in this same SCI model that the use of MLIF in combination with other neuroprotective peptides, such as A91 and GSHE, contributes to promote a better neuroprotective effect by preserving the medullary parenchyma and axonal fibers. The combined strategy increased the number of motor neurons and decreased the presence of collagen, promoting a better motor recovery in rats after SCI [124].

##### Use of A91 Immunization Combined with Cell Therapy

Immunization with A91 has been shown to provide neuroprotection in SCI [125,126]. However, the neuroprotective effect achieved by the immunization strategy is related to the genetic background of the animals (strains), the type of adjuvant, and the intensity of the lesion [127], important characteristics that could be considered for the use of combination therapies.

Dendritic cell (DC) strategies help reduce the required antigen dose, while further limiting the cross-presentation of other cell types for their antigen-presenting ability. For the stimulation of an immune response, DC migrates to the spleen or draining lymph nodes. There, DCs mount the immune response to antigens and can also mount the autoantigen-specific responses seen in autoimmunity [128,129]. Immunization utilizing dendritic cells pulsed with a combination of spinal cord homogenate (hpDC) and A91 was reported to increase levels of BDNF, NT-3, IL-4, and IFN-γ at the injured site, as well as in the supernatant of cultured T cells of the hpDC group, which was significantly higher than that of the control groups [130] and promoted functional motor recovery [131]. A91-pulsed dendritic cells improved the levels of BDNF and NT-3 expression and exerted a neuroprotective effect and possible regeneration in an SCI mouse model [132]. Evidence has shown that a high concentration of neurotrophic factors in the SCI is beneficial for axonal growth and neuronal survival [133,134]. Under pathological conditions, T cells are a source of neurotrophic factors, and the release of these molecules can be significantly increased by antigen activation [135]. Some CNS growth factors have been reported to be continuously produced and released at the site of injury by activated T cells [136,137]. Further studies are needed to test this strategy in SCI rat models to compare the beneficial effect with another combination.

In the chronic phases, cell necrosis, glial reaction, and inflammation induce the appearance of glial cavities, cysts, and scars that interrupt the descending and ascending axonal tracts, preventing possible axonal regeneration [131]. In the chronic phase of SCI, there is scar tissue that prevents the correct reconnection of axons by forming a physical and chemical barrier of sulfated proteoglycans, thus inhibiting the formation of growth cones and axonal prolongation [6,138,139,140]. Additionally, chronic SCI is considered as a period of low activity with progressive decline, so a possible alternative to achieve axonal regeneration could be scar glial removal (SGR), with the aim of restoring axonal connections and, consequently, synapses in addition to restoring the conditions of an acute injury, such as the activation of protective autoimmunity, as well as the release of cytokines and neurotrophic factors [131,137,141].

Previous studies have demonstrated that SGR or implanting bone marrow mesenchymal stem cells separately leads to significant tissue restoration and improved motor recovery following SCI [55,130]. Building upon these findings, the potential enhancement of therapy by combining A91 immunization with these alternative strategies was explored. In an initial approach, the impact of A91-peptide immunization alone and in conjunction with SGR was investigated regarding locomotor recovery, gene expression related to regeneration, cytokine levels, and the count of regenerating axons in a chronic SCI model. The results indicated that both treatments could substantially modify the non-permissive microenvironment characteristics of the chronic phase of SCI, thereby creating an environment conducive to greater motor recovery [132].

Subsequently, the investigation was directed to design a combination therapy that integrated a fibrin–glue matrix-like scaffold combined with mesenchymal stem cells, and A91 immunization 72 h after SCI or in the chronic phase (60 days after treatment). The intervention, in acute phases (72 h), of the combination strategy after moderate SCI was the best strategy to promote motor and sensibility recovery at 60 days after treatment. In addition, significant increases in tissue preservation and axonal density were observed, suggesting that these therapies exhibit potential effects on the protection and regeneration of neural tissue after acute spinal cord injury [142]. In chronic phases after a complete SCI transection model, the combination therapy promoted motor and electrophysiological recoveries. In the same way, the combined therapy was capable of inducing a permissive microenvironment for neuroregeneration [118].

In a subsequent study, the same combination was also evaluated in the chronic phase but in an SCI contusion model. The outcomes indicated that the combined therapy had the capacity to alter the non-permissive microenvironment following SCI. However, it fell short of inducing substantial axonal regeneration or neurogenesis, contrasting the outcomes observed after exclusive A91-immunization treatment [132].

A91 immunization combined with other therapeutic strategies could be useful for other acute or chronic neurodegenerative and neuroinflammatory pathologies. However, at present, there is a lack of evidence on the beneficial effects of A91 for pathologies other than SCI. The chronology of A91-immunization research is summarized in Table 3.

The effectiveness of any therapeutic strategy also depends on other factors, such as the nutritional status and balance of intestinal microbiota. These elements could also determine the permissible microenvironment that will favor the protection and/or restoration of tissue after SCI. It is worth mentioning that the CNS, due to its metabolic role, added to the impossibility for controlling the inflammatory response and would require other external agents that would promote changes in the microbiota and their metabolites involved in neuroprotective and neurodegenerative processes. Various strategies have implemented the use of supplements and probiotics in animal models and patients, which are described in Section 4.

## 4. Supplements and Probiotics as Therapeutic Strategies after Spinal Cord Injury Inflammation

### 4.1. Supplements

The European Food Safety Authority (EFSA) defines food supplements as a “concentrated source of nutrients or other substances with nutritional or physiological effect that are marketed in dose form. A wide range of nutrients and other ingredients may be present in food supplements, including, but not limited to, vitamins, minerals, amino acids, essential fatty acids, fiber, and various plant and herbal extracts” [150].

Dietary supplements have been proposed for various neurodegenerative diseases and post-traumatic conditions, as they have been studied for their anti-inflammatory and neuroprotective effects on the brain.

Supplement use is common in people with SCI, with multivitamins, calcium, and vitamin D being the most commonly used [151]. Currently, supplementation is also being tested to reduce inflammation induced by SCI. There are several animal and human reports exploring the beneficial effects of various supplements.

#### 4.1.1. Animal Studies

##### Vitamins

Vitamin C (ascorbic acid) is a water-soluble vitamin and an essential nutrient found in high concentrations in the brain. It is well known for its antioxidant role in the brain and its catalytic effect on various enzymatic reactions [152].

Intracellular vitamin C plays important roles in the central nervous system, such as myelin formation, neurotransmission modulation, and antioxidant protection [153]. A part of the protective action of this vitamin in the CNS is because it promotes the removal of glutamate from the synaptic cleft and inhibits calcium channels. Regarding neuroinflammation, vitamin C reduces the production of inflammatory cytokines, such as IL-β and TNF-α, which, in turn, increases glutamatergic neurotransmission [154].

In a comparative study evaluating inflammatory markers in rats after SCI, serum levels of TNF-alpha, IL-6, and nitrite were significantly reduced compared with the control group with the administration of 100 mg/kg of ascorbic acid for 45 consecutive days [155].

Vitamin E (α-tocopherol) is known as the main fat-soluble antioxidant in the body. Like vitamin C, humans must obtain vitamin E through the diet because we are not capable of biosynthesizing it [156]. Vitamin E has the ability to reduce the neuroinflammatory responses of microglial cells, increase the number of oligodendrocytes [157], and decrease IL-1beta and TNF-alpha levels in the hippocampus of animal models [158]. After vitamin E supplementation for 14 days after SCI, there was an improvement in BBB scores (Basso, Beattie, and Bresnahan scale), acting as an inhibitor of lipid peroxidation products, such as thiobarbituric acid reactive substances and malondialdehyde [158]. The anti-inflammatory actions of vitamins C and E as antioxidants have been shown in several animal studies to lead to improved recovery of motor function after SCI. Although oral intake was not associated with a significantly different effect compared to no treatment, favorable results were observed with intraperitoneal administration of vitamin C. Simultaneous administration of combined vitamins C and E did not provide any additional therapeutic effect. The pharmacological action of these vitamins against free radicals and oxidative stress makes them potential candidates for the pharmacological treatment of SCI, possibly limiting damage in the first phase and inhibiting the spread of damage during the second phase [154].

Supplementation with trace elements and vitamin E maintains the number and function of T-lymphocytes after acute SCI. This therapeutic strategy also promotes motor recovery, which is probably due to its antioxidant effect [159].

##### Minerals

Zinc is present as a cofactor in more than 300 enzymes and is involved in remodeling mechanisms after injury in many types of tissue. Copper/zinc-dependent superoxide dismutase decreases oxidative stress and attenuates the blood–brain barrier mediated by MMP9 matrix metalloproteinases. This is important because ischemic and damaged neuronal tissues are affected by the lipid peroxidation of the cell membrane, leading to edema and local inflammation [160]. Adequate zinc concentrations can inhibit inflammation, promote autophagy, inhibit oxidative stress, protect injured neurons, and treat spinal cord injuries. However, if administered at high concentrations, zinc can impair neurons through excitotoxicity, inducing oxidative stress and impairing cellular energy production. Therefore, the appropriate concentration of zinc could be used in SCI, but this dose has yet to be further defined [161].

In a rat study [162], subjects were administered 30 mg/kg of zinc in an intraperitoneal manner. The researchers found that it promoted the recovery of SCI by decreasing neuroinflammation in vitro and in vivo by reducing IL-1beta, casp1, and NLRP3 inflammasome levels [162].

##### Trace Elements

Certain trace elements are involved in development and regeneration processes [160].

Selenium is an essential trace element, which participates in the structure of glutathione peroxidase and is widely known for its harmful effects in case of deficiency [163]. The selenocysteine (sec)-containing group of selenoproteins contributes to protecting cells from oxidative damage, especially through the selenium-dependent families of glutathione peroxidases and thioredoxin reductases. Nanoparticles in rats with SCI have a neuroprotective effect and accelerate neuronal function by controlling the inflammatory response [164]. The essentiality of the selenoproteins GPX4, thioredoxin reductase 1 (TXNRD1), and thioredoxin reductase 2 (TXNRD2) for the normal development and survival of neurons became well understood after studying the phenotypes of respective gene knockout mice, which resulted in failed embryonic development [165].

Copper is an essential trace element and an important catalyst for heme synthesis and iron absorption. After zinc and iron, copper is the third most abundant trace element in the body [166]. It serves as a catalytic factor in redox chemistry for biologically active proteins (e.g., superoxide dismutase, cytochrome c oxidase, and lysyl oxidase). All these proteins function to regulate growth, metabolism, and oxidative stress, aiding in the pathophysiology of SCI and possible neuroregeneration after injury [167].

Seelig’s group, in their 2019 study, suggested a potentially favorable association between the use of copper and selenium in damaged cells and the potential for a better anti-inflammatory response, which also allows a more reliable prognosis after SCI [167].

##### Omega-3 Fatty Acids

Polyunsaturated fatty acids (PUFAs) include α-linolenic acid (ALA), stearidonic acid (SDA), eicosapentaenoic acid (EPA), docosapentaenoic acid (DPA), and docosahexaenoic acid (DHA) oils that contain these fatty acids (FAs). These FAs originate mainly from certain plant, algal, and unicellular sources [168]. Essential FAs have been reported to be important in membrane fluidity, inflammatory eicosanoids, and neural membrane oxidation [169].

A mouse model study by Marinelli and collaborators confirmed and extended the beneficial effects of repeated systemic administration of DHA in the primary and secondary phases of SCI. DHA treatment was shown to improve motor and sensory functions, exert neuroprotective actions, modulate cell responses to injury (apoptosis and survival) and inflammation in astrocytes and microglia, and, ultimately, promote spinal regeneration. In recent years, there has been a debate about whether the action of omega-3 fatty acids, mainly DHA, in neurodegenerative disorders exerts anti-inflammatory, neuroprotective, and pro-regenerative actions. In line with this, it has already been shown that DHA is capable of exerting an anti-inflammatory effect not only in the acute phase of SCI but also in the intermediate/chronic phases of SCI [170].

In SCI, the apoptosis of oligodendrocytes and neurons leads to increased oxidative stress and inflammation. A common explanation for the anti-inflammatory effects of omega-3 PUFAS is that they compete with arachidonic acid (AA), the main representative of the omega-6 family of PUFAs. AA is the precursor of prostaglandins, thromboxanes, and leukotrienes, which have proinflammatory effects [171].

##### Antioxidants

Oxygen-radical-induced lipid peroxidation is perhaps the most important deleterious phenomenon that develops owing to inflammation after SCI. Therefore, various therapeutic strategies are aimed at neutralizing the harmful effects of this process [172]. Resveratrol is a naturally occurring polyphenol found in grapes and is effective in preventing redox spoilage. It has anti-inflammatory properties by altering arachidonic acid metabolism and inhibiting protein kinase, making it a possible therapeutic strategy in SCI [172].

Senturk and Zhao demonstrated, in two different studies in rat models, that the application of intraperitoneal resveratrol after SCI (10 mg/kg and 30 mg/kg, respectively) decreased the levels of proinflammatory cytokines IL-1 beta, IL-6, and TNF-alpha, which promoted better motor recovery [173,174].

##### Botanicals

Beta-carotene is a tetraterpenoid consisting of a C40 structure that includes two β-ionone rings. Along with lycopene, it is among the most frequently consumed dietary carotenoids in humans and has among the highest plasma concentrations [175]. Beta-carotene could improve functional recovery and spinal histological changes, as well as inhibit oxidative stress and inflammation by inhibiting NF-KB in the spinal cord of SCI rats. Further clinical experiments or human studies are suggested [176].

Spirulina platensis is a microalga that is most often used as a supplement. It contains carotenoids, phycocyanin, xanthophylls, and phycobilins, which have antioxidant activities [177]. Spirulina platensis supplementation after the induction of SCI significantly improves functional recovery. Significant ultrastructural improvement was also observed with reduced progression of morphological damage by preserving the ultrastructure of the spinal cord from secondary injury. This suggests that Spirulina platensis could be used in SCI patients to induce functional recovery [1].

#### 4.1.2. Human Studies

##### Vitamins

Vitamin D is a generic term because it refers to a group of fat-soluble compounds with a main chain comprising cholesterol rings [178]. Despite our growing understanding of the extraskeletal functions of vitamin D, this vitamin has been poorly studied in SCI. Low calcifediol (25-OHD) levels might be a marker of the severity of the inflammatory process rather than an accurate reflection of vitamin D status. Shortly after injury, people with SCI show acute inflammation because of trauma, which then becomes chronic. Although the impact of chronic inflammation on vitamin D status has never been reported, the interpretation of 25-OHD levels in an SCI setting should consider the inflammatory status of the patient [179].

##### Omega-3 Fatty Acids

The use of Omega-3 PUFAs could suppress the production of proinflammatory cytokines, such as IL-6, IL-1β, and TNF-α, in human models in addition to suppressing the production of bone absorption in patients with SCI; however, until now, there has not been enough conclusive information available for patients with SCI. The group of Sabour , in their study, demonstrated inconsistency in the reported effects of Omega-3 PUFAs on proinflammatory cytokines, reporting that no benefit was found in reducing the levels of these cytokines [180].

Omega-3 polyunsaturated fatty acids have anti-inflammatory effects. In SCI patients the effects of therapy with omega-3 polyunsaturated fatty acids were investigated and levels of plasma leptin and adiponectin were evaluated. This study concluded that leptin concentrations were not influenced; however, adiponectin decreased significantly [180]. It has been reported that PUFAs exert a neuroprotective effect and conducted a double-blind study in patients with SCI, where the authors evaluated the influence of omega-3 PUFA consumption on neurorehabilitation in patients with SCI by assessing disability. It was reported that Omega-3 fatty acids have a neuroprotective effect in the acute phase of SCI but not in the chronic phase [181].

The key characteristics of these supplements used in human studies are summarized in Table 4.

### 4.2. Probiotics

The intestinal flora includes a heterogeneous group of bacteria that live in the gastrointestinal system and influence human health, especially through interactions with the immune system, skin, lungs, and brain. The microbiome–gut–brain axis is a well-established research term that includes connections of afferent and efferent neurons and endocrine and metabolic signaling that make the brain and gut connected. The modulation of neurotransmission in gut dysbiosis has been postulated to play a key role in the pathogenesis of CNS disorders. An alteration in the intestinal microbiome causes an increase in proinflammatory molecules, which translates to an alteration in the permeability of the blood–brain barrier. Spinal cord injuries disrupt the autonomic nervous system; they also cause dysbiosis, which is essentially an imbalance in the gastrointestinal tract that affects the gut microbiota [182].

Probiotics, which are regulated as dietary supplements and foods, consist of yeasts or bacteria. They come in various presentations, such as powders, tablets, and capsules, and in fermented foods, dairy products, or drinks, such as kombucha. Probiotics are available as a single microorganism or as a mixture of several species [183].

The use of probiotics as therapeutic strategies after SCI is also a topic under study.

#### 4.2.1. Animal Studies

An animal study demonstrated that SCI in mice affected gastric permeability and induced gut dysbiosis, which, in turn, led to delayed locomotor recovery. Therapeutic management with commercially available probiotics resulted in improved locomotor recovery [184,185].

Recent data from rodents indicate that SCI causes gut dysbiosis that exacerbates intraspinal inflammation and injury pathology, leading to poor recovery of motor function. Post-injury administration of probiotics containing various types of beneficial bacteria may partially overcome the pathophysiological effects of gut dysbiosis. The immune function, locomotor recovery, and spinal cord integrity are partially restored by a sustained regimen of oral probiotics. Recently, in a mouse model, Phylum Bacteroidetes and Phylum Firmicutes, the two major bacterial orders in the gut, were revealed to be inversely regulated by SCI three weeks after injury. Phylum Bacteroidetes levels decreased by 30%, and Phylum Firmicutes levels increased by 250% relative to pre-injury values. Given that the bacterial orders Bacteroidales and Clostridiales together constitute >80% of all the species residing in the gut, significant and long-lasting changes in their relative population densities after SCI are likely to influence numerous physiological processes. For instance, communication between the gut microbiota and gut-associated lymphoid tissue (GALT) immune cells produces cytokines and other metabolites that circulate and affect CNS functions. Dysbiosis is associated with marked changes in the relative proportion of immune cells found in mesenteric lymph nodes and Peyer’s patches. There is also an increase in the synthesis of inflammatory and immunoregulatory cytokines in GALT in parallel with changes in immune cell populations. Gut microbes also produce neuroactive metabolites (short-chain fatty acids) and neurotransmitters, which can affect central nervous system functions by activating vagal afferent nerve fibers in the gut. There is scarce information available about whether probiotics can confer neuroprotection or ameliorate various comorbidities and neurological complications caused by traumatic SCI [184,186,187].

Studies that have sought to correct the gastrointestinal microbiota (GIM) have seen a decrease in the levels of inflammatory cytokines. The GIM also plays an essential role in the production of neurotransmitters. Lactobacillus and Bifidobacterium, two genera of bacteria for which levels decline after injury, play important roles in the production of neurotransmitters, such as serotonin, dopamine, and *γ*-aminobutyric acid [188].

Gut microbiota depletion significantly reduced glial cell line-derived neurotrophic factor (GDNF) expression in mice. Mice deficient in TLR-2 had increased susceptibility to inflammation. This highlights the importance of the gut microbiota-TLR-2-GDNF axis in modulating the enteral nervous system and inflammation [189].

In a study conducted by Lin and collaborators, involving rats, the timely administration of conditioned medium from Lactobacillus rhamnoides GG (LGG-CM) shortly after SCI yielded notable reductions in the extent of post-traumatic inflammation in proximity of the injury site. Additionally, this treatment facilitated the recovery of locomotor function subsequent to the SCI. This beneficial effect of LGG-CM can be attributed to its inhibition of the NK-*κ*B pathway, which is achieved through the reduction of I*κ*Bⲁ phosphorylation. This inhibition prompts a shift in microglia/macrophage polarization toward the M2 phenotype and simultaneously curtails polarization toward the M1 phenotype. Consequently, LGG-CM presents itself as a potential therapeutic adjunct for promoting neuroprotection following an SCI event [173].

#### 4.2.2. Studies in Humans

A clinical study was conducted to show that the gut microbiome in SCI patients was different compared to that in healthy adults. They tested 30 SCI patients with different types of bowel dysfunction and 10 healthy controls. Microbial patterns were taken from stool samples. Firmicutes and Bacteroides spp are the predominant phyla in the intestine. They ferment indigestible polysaccharides and generate metabolites that the host can use for energy. Among them, butyrate is the most pronounced single-chain fatty acid that has modulatory effects on epithelial cell growth and differentiation and immune functions and a potent anti-inflammatory effect on macrophages and suppresses ongoing inflammation in the CNS. In this study, there was a significant reduction in butyrate levels in SCI patients, suggesting that reduced butyrate levels may contribute to microglia-mediated neurotoxicity in these patients and implying that low butyrate levels may have an impact on long-term recovery after SCI. These results have been supported by a series of studies where the levels of butyrogenic species are reduced after SCI, and this persists in chronic stages [190,191]. Therefore, supplementation with butyrate-increasing probiotics could be a therapeutic strategy for SCI. The key characteristics of this probiotic used at the clinical level are summarized in Table 4.

**Table 4 ijms-24-13946-t004:** Clinical studies of supplements or probiotics used after SCI.

Interventions	Sample Sizes	Study Designs	Results/Findings	Medical Risks	References
Vitamin D	Total of 34 patients	Supplementation	About 62% of participants improved handgrip strength postsupplementation	No medical risks	[192]
Omega-3 fatty acids	Total of 104 patients with SCI	Double-blinded randomized clinical trial	The data showed that omega-3 fatty acids may not affect plasma concentrations of leptin but adiponectin level was decreased in patients with SCI	No medical risks	[180]
Polyunsaturated fatty acids	Total of 110 patients	Double-blinded randomized clinical trial	No changes were observed in either group with the consumption of ω-3 fatty acids	No medical risks	[181]
n-3 fatty acids	Total of 75 patients	Double-blinded, placebo-controlled trial	Neither the supplemented nor control groups showed any difference in their baseline characteristics. There were no significant differences between both groups at the end of the study or in each group between the beginning and end of the study.	No medical risks	[180]
Probiotics	207 eligible participants with SCI and stable neurogenic bladder management	Multi-site randomized, double-blinded, double-dummy, placebo-controlled trial	There was no effect of RC14-GR1 or LGG-BB12 in preventing urinary tract infections in people with SCI.	No medical risks	[193]

## 5. Conclusions

Spinal cord injury continues to be a health problem for which there is no fully effective treatment. Pharmacological and non-pharmacological strategies have been evaluated and have provided different expectations to improve the neurological recovery of injured individuals. Nevertheless, owing to the complexity of the events generated after injury—particularly the one presented by the inflammatory response—it has not been easy to establish an effective therapy that may help neuroprotection or promote neuronal regeneration. The approach for modulating the inflammatory response has been a difficult topic of investigation. The use of cell therapy, immunomodulatory peptides or even various supplements as modulators of the immune response has raised hopes of finding the best strategy to neutralize the harmful effects of inflammation. However, the road to the final goal is still long and requires more research to obtain better results at a basic leel that can be extended to clinical applications.

## Figures and Tables

**Figure 1 ijms-24-13946-f001:**
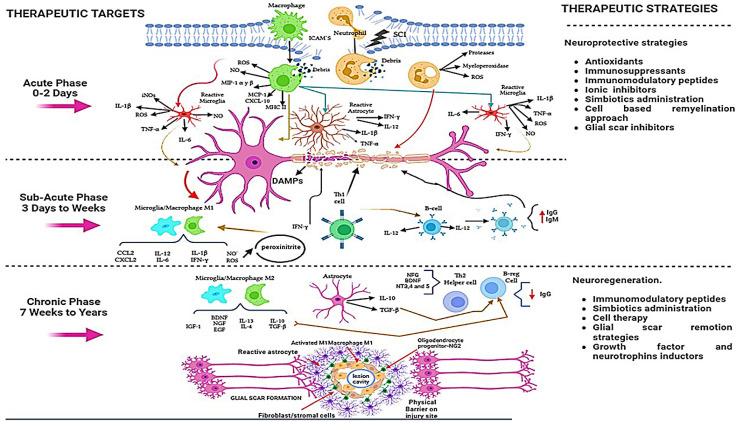
Self-destructive mechanisms and therapeutic targets in different phases of SCI. In the acute and subacute phases, resident and peripheral cells determine neural destruction through proinflammatory cytokines (IL-1β, IL-6, TNF-α, IFN-γ, and IL-12), chemokines (MCP-1, CXCL10, CCL2, and CXCL2), free radicals (ROS and NO produced by iNOS), and other molecules (proteases, DAMPs, IgG, and IgM). For this reason, it is imperative to develop strategies that act on different molecular mechanisms for reducing damage. In the chronic phase, the therapeutic strategies should be directed to limit the neurodegenerative process and promote a favorable microenvironment for neural restoration. In line with this, glial scar removal and the release of anti-inflammatory cytokines (IL-10, IL-13, IL-4, and TGF-β), growth factors (BDNF, NGF, EGF, and IGF-1), and neurotrophins (NT3, 4, and 5) could play a significant role. Recently, dual effects (detrimental and beneficial) of neuroinflammation have been reported [4,8]. Inflammation is a process of enormous complexity, with some mechanisms detrimentally contributing to further secondary damage and others—if modulated—contributing beneficially to the protection and restoration of SCI [9,10]. For this reason, diverse therapeutic strategies are directed to modulate, or even inhibit, this inflammatory response.

**Table 1 ijms-24-13946-t001:** Immune responses related to cell types used in SCI therapy.

Cells	Immune Responses	Effect of SCI	References
SCs	Modulation of macrophages from M1 to M2 Reduced levels of NLRP3, TNF-α, and IL-1β Produce NGF, BDNF, CNTF, and NT-3	Neuroprotection and neuroregeneration	[18,19,21,28,29]
OECS	Modulation of the inflammation regulated by IL-10 and TGF-b Promote phagocytosis via integrin receptors Induce the polarization M1-M2 by inhibiting JAK/STAT3- Decreased TNF-α, IL-1β, and oxidative stress levels Increased IL-10 levels Modulated responses A1 to A2	Neuroprotection	[32,35,39,40]
NG2/OPCs	Upregulation of CXCR4 and CXCL12 in astrocytes Increase levels of IL-17, IL-6, and metalloproteinases MMP3, MMP9	Neuroprotection	[44,46]
BMSCs	Modulation from M1 to M2 after SCI Decrease levels of TNF-α, IL-1β, and IL-6 Increase expression of IL-4, IL-10, IL-13, and TGF-β Inhibit T-cell activation by TGF-β	Neuroprotection and neuroregeneration	[16,57,58,59,60]
NSCs	Secrete BDNF, CNTF, GDNF, NGF, and IGF-1 Regulation of T-cells and macrophages Reduce glutamate exposure and levels of pro-apoptotic markers Increase levels of anti-apoptotic markers	Neuroprotection	[61,62,63]
Aldynoglia	Promote polarization from M1 to M2 Upregulation to SIRT-2	Neuroprotection and neuroregeneration	[64,65]

Scs, Schwann Cells; OECs, Olfactory Ensheathing Cells; NG2/OPCs, Oligodendrocyte Precursor Cells; BMSCs, Bone-Marrow-Derived Mesenchymal Stromal Cells; NSCs, Neural Stem Cells.

**Table 2 ijms-24-13946-t002:** Clinical studies using different cell types for SCI therapy.

Cell Type	SCI Level	Dose	Outcomes	Adverse Events Post-Transplant	References
SCs	Chronic; T6-T9 ASIA A or C	3 × 10^6^ or 4.5 × 10^6^ cells in 300 µL	No motor or sensory improvement	No adverse effects	[72]
SCs	Thoracic or cervical ASIA A or B	3 × 10^6^ cells in 300 µL	They showed improvement in touch; some of the patients showed improvement in sphincter and had an increase in FIM and FAM scores.	No adverse effects	[66]
SCs	C5-T12; ASIA A-C	4 × 10^6^ or 6 × 10^6^ cells in 200 µL	They showed an increase in ASIA and FIM scores and an increase in period latencies and wave amplitude in SSEPs and MEPs.	No adverse effects	[73]
SCs	Sub-acute; ASIA A	5 × 10^6^, 10 × 10^6^, and 15 × 10^6^ cells	The patients showed an improvement in FIM.	No adverse effects	[74]
Autologous Olfactory Ensheathing Cells	Chronic; thoracic paraplegia ASIA A	3 × 10^4^ and 2 × 10^5^ cells	Transplantation was safe and feasible. The first two operated patients improved from ASIA A to ASIA C and ASIA B.	No adverse effects	[68]
Autologous Olfactory Ensheathing Cells	Chronic; cervical paraplegia A, B, and C	1 × 10^6^ cells in 1 mL	Return of substantial sensation and motor activity in various muscles below the injury level was observed in three patients. In addition, bladder function was restored in two patients.	No adverse effects	[69]
Autologous Olfactory Ensheathing Cells	Chronic thoracic paraplegia ASIA A	12 × 10^6^, 24 × 10^6^, and 28 × 10^6^ cells	There were no significant functional changes in any patients and no neuropathic pain. In one patient, improvement in three segments in light touch and pin prick sensitivity bilaterally, anteriorly, and posteriorly.	No adverse effects	[75]
Olfactory mucosa autografts	Chronic; C4-T6 ASIA A		Two patients reported return of sensation in their bladders, and one of these patients regained voluntary contraction of anal sphincter. Two of the seven ASIA A patients became ASIA C.	No adverse effects	[76]
Fetal olfactory ensheathing glia cells	Complete injury; thoracic and cervical level	5 × 10 ^5^ cells in 0.05 mL	There were no significant functional changes in any patients and no neuropathic pain. In one patient, improvement in three segments in light touch and pin prick sensitivity bilaterally, anteriorly, and posteriorly.	No adverse effects	[77]
OPCs	Sub-acute; C4-7 ASIA A or B	2 × 10^6^, 1 × 10^7^, Or 2 × 10^7^ cells	Thirty-two percent of patients recovered two or more points in neurological functions on at least one side of the body, and 96% improved one or more points.	No adverse effects	[70]
BMSCs	C7-T11; ASIA A	10 × 10^6^ cells in 2 µL	Increase in ASIA scores; no motor score improvement.	No adverse effects	[78]
BMSCs	Chronic; ASIA A	4 × 10^6^ cells	Patients had a 58% recovery rate in ASIA.	No adverse effects	[79]
BMMNC	Sub-acute; ASIA A: 4 or B: 1	1 × 10^10^ cells	The patients showed 40% improvement based on the ASIA scale.	No adverse effects	[80]
BMSCs	Chronic; ASIA A	2 × 10^7^ cells	Showed 45% improvement in ASIA of A and B.	No adverse effects	[81]
AMSCs	Chronic; ASIA A and B	4 × 10^8^ cells	They had 12.5% improvement in ASIA.	No adverse effects	[82]
BMMNCs	Chronic; ASIA C	1 × 10^10^ cells	Showed 0% improvement in the ASIA; only showed improvement in SEP/MEP.	No adverse effects	[80]
BMSCs	Chronic; complete injury	120 × 10^6^ cells	They showed an improvement in clinical aspects and in the quality of life of the patients.	No adverse effects	[83]
Hu-NSCs	Chronic; ASIA A and B	2 × 10^8^ and 4 × 10^8^ cells	They showed improvement in motor function.	No adverse effects	[84]
NSCs	Sub-acute; ASIA A	1.2 × 10^6^ cells	Showed improvement in EMG electromyography.	No adverse effects	[85]
hNSCs	ASIA A or B cervical injuries were sensorimotor complete	1 × 10^5^ cells in 1 mL	Patients showed moderate improvement based on the ASIA scale.	No adverse effects	[86]

ASIA, American Spinal Injury Association; FIM, Functional Independence Measure; FAM, Functional Assessment Measure; SSEPs, Somatosensory Evoked Potentials; MEPs, Motor Evoked Potentials; AMSCs, Adipocyte Tissue-Derived Mesenchymal stromal cells; BMSCs, Bone-Marrow-Derived Mesenchymal Stromal Cells; BMMNCs, Bone-Marrow-Derived Mononuclear cells; EMG, Electromyography; NSCs, Neural Stem Cells; OPCs Oligodendrocyte Precursor Cells.

**Table 3 ijms-24-13946-t003:** Chronology of A91-immunization results after SCI.

**A91 Immunization Alone**	**Injury Models**	**Doses**	**References**
Compared with methylprednisolone, showed improved motor recovery and increase in number of rubrospinal neurons	MC	Methylprednisolone (30 mg/kg, IV) and A91 (150 µg, ID) emulsified with complete Freund’s adjuvant (0.5 mg/mL)	[98]
A91 immunization reduced lipid peroxidation levels in contusion model	MC	A91 (150µg, ID) emulsified with complete Freund’s adjuvant (0.5 mg/mL)	[143]
In MC or IT, showed significant anti-A91 T cell proliferation and increased IL-4 and BDNF production, which were different in SC or CT	MC, SC, IT, and CT	A91 (150 µg, ID) emulsified with complete Freund’s adjuvant (0.5 mg/mL)	[117]
Immunization with A91 or Cop-1 reduced NO production and iNOS gene expression in rat and mouse SCI models	MC and C	A91 (150 µg, ID) and Cop-1 emulsified with complete Freund’s adjuvant (0.5 mg/mL)	[117]
The induction of immunological tolerance to A91 at birth resulted in substantial motor recovery and enhanced survival of rubrospinal and ventral horn neurons	MC	A91 (75 µg, ID) alone at 45 years old and another group with a booster dose (75 μg, ID) 24 h after the first dose emulsified with complete Freund’s adjuvant (0.25 mg/mL)	[125]
Reduced apoptosis caused by SCI by decreasing Casp3 activity and TNF-α levels	MC	A91 (150 µg, ID) emulsified with complete Freund’s adjuvant (0.5 mg/mL)	[126]
Induced a long-term production of BDNF and NT-3, leading to improvements in motor recovery during the chronic stages	MC	A91 (150 µg, ID) emulsified with complete Freund’s adjuvant (0.5 mg/mL)	[144]
A91 or Cop-1 significantly reduced IL6, IL1β, and TNFα and increased IL10, IL4, and IGF-1 gene expressions in MC in contrast to SC	MC and SC	A91 (150 µg, ID) and Cop-1 (150 μg, ID) emulsified with complete Freund’s adjuvant (0.5 mg/mL)	[145]
**A91 Immunization with Other Strategies**	**Injury Models**	**Doses**	**References**
A91 plus GSHE induced better motor recovery and increased numbers of myelinated axons and rubrospinal neurons compared to A91 alone	MC	A91 (150 µg, ID) emulsified with complete Freund’s adjuvant (0.5 mg/mL)	[146]
A91 plus GSHE induced better motor recovery and an increase in the number of rubrospinal and ventral horn neurons when GSHE was applied immediately after or within the first 72 h after the injury	MC	A91 (150 µg, ID) emulsified with complete Freund’s adjuvant (0.5 mg/mL) GSHE total dose (12 mg/kg, IP) divided into four injections (20 min and 4, 10, and 20 h) post-injury	[147]
A91 combined with MLIF and GSHE induced a better preservation of parenchyma and axonal fibers, increased the number of motor neurons, and reduced the amount of collagen	MC	A91 (150 µg, ID) emulsified with complete Freund’s adjuvant (0.5 mg/mL) GSHE total dose (12 mg/kg, IP) divided into four injections (20 min and 4, 10, and 20 h) post-injury. MLIF (4 µg) divided into four injections, immediately on injury site and then IP every 24 h	[124]
**A91 Immunization with** **Other Regeneration Strategies**	**Injury Models**	**Doses**	**References**
DCs stimulated with A91 enhanced the expression levels of BDNF and NT-3, exerting a neuroprotective effect and potentially promoting regeneration in a mouse model.	C	DCs (2 × 10 ^6^ cells/mL) stimulated with A91 (100 mg/mL) in the medium were collected and injected IP (1 × 10 ^6^ cells/0.3 mL) 24 h after SCI	[132]
Compared with A91 alone, A91 in combination with SGR made it possible to modify the non-permissive microenvironment during the chronic phase, thereby offering an opportunity to enhance the motor recovery.	MC	A91 (200 µg, ID) emulsified with complete Freund’s adjuvant (0.5 mg/mL) two months after injury and SGR	[148]
The combination of SGR, fibrin matrix, MSCs, and A91 was demonstrated to be the most effective approach for enhancing motor and sensory recoveries, preserving tissue, and increasing axonal density in acute phases	MC	A91 (150 µg, ID) emulsified with complete Freund’s adjuvant (0.5 mg/mL) A mixture of MSCs (2.5 × 10^6^ cells in 5 µL) and FG (10 µL) was grafted at the site of the injury.	[142]
The combination of SGR, fibrin matrix, MSCs, and A91 promoted motor and electrophysiological recoveries in the chronic phase	CT	A91 (150 µg, ID) emulsified with complete Freund’s adjuvant (0.5 mg/mL) A mixture of MSCs (2.5 × 10^6^ cells in 5 µL) and FG (10 µL) was grafted at the site of the injury two months after the injury.	[145]
The combination of SGR, fibrin matrix, MSCs, and A91 modified the non-permissive microenvironment post SCI, but it was not capable of inducing an appropriate axonal regeneration or neurogenesis compared to the treatment with A91 alone	MC	A91 (150 µg, ID) emulsified with complete Freund’s adjuvant (0.5 mg/mL) A mixture of MSCs (2.5 × 10^6^ cells in 5 µL) and FG (10 µL) was grafted at the site of the injury, two months after the injury.	[149]

MC, moderate contusion; IT, incomplete transection; SC, severe contusion; CT, complete transection; C, compression; IV, intravenous; ID, intradermal; IP, intraperitoneal; SGR, scar glial removal; DCs, dendritic cells; MSCs, mesenchymal stem cells.

## Data Availability

Not applicable.

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
