# Peer review of "Use of Cells, Supplements, and Peptides as Therapeutic Strategies for Modulating Inflammation after Spinal Cord Injury: An Update"

_ijms, 2023, doi:10.3390/ijms241813946_

Round 1

Reviewer 1 Report

This manuscript entitled “Use of cells, supplements and peptides as therapeutic strategies for modulating inflammation after spinal cord injury: An update” has been reviewed sincerely. Authors considered the therapeutic candidates of spinal cord injury using suitable disease models. I felt the topic is interesting. Although I have no serious comments in this submitted review paper, I described several points for further improvement.

1.        Present clinical therapies for SCI should be summarized in “Introduction” section.

2.        Animal studies and human studies should be distinguished throughout paper. Especially, new Table containing sample size, study designs, results/findings of the clinical study should be provided. The Table will help the understanding of the readers.

3.        The possibility of medical risk by the presented therapy should be discussed in parallel.

Author Response

Answering to Reviewer 1

Query 1. Present clinical therapies for SCI should be summarized in “Introduction” section.

Answer: We appreciate this observation and have added the requested information. (Page 5 and 6 Lines 106 to 129)

Query 2. Animal studies and human studies should be distinguished throughout paper. Especially, new Table containing sample size, study designs, results/findings of the clinical study should be provided. The Table will help the understanding of the readers.

Answer: We have already distinguished between animal and human studies in the sections related to cell therapies and supplement and probiotic strategies. Regarding the section related to immunomodulatory peptides there are not clinical studies, thereby, we only provide animal studies.  We have also included two new tables (tables 2 and 4) and have modified tables 1 and 3.

Query 3. The possibility of medical risk by the presented therapy should be discussed in parallel.

Answer: We have included this information in the corresponding table.

Regards, 

Authors

Reviewer 2 Report

This review summarizes the cells, supplements and peptides that are useful for spinal cord injury, focusing on inflammatory reactions. Although a variety of research results are presented in a comprehensive manner, we believe that the following revisions will make the paper more meaningful to readers. Especially, it would be desirable to make the tables easier for the reader to understand.

It is necessary to be improved as there is little mention of the method and timing of administration, which is important in the treatment of spinal cord injury.

Various cell sources useful for spinal cord injury have been clinically researched in humans. For each of the cells presented by the authors, it should be clearly stated to what extent the animal experiments and clinical trials have revealed what is known about the cells. In this point, Table 1 is not clear, which should be corrected. Table 2 also contains long sentences, which are difficult to understand for the reader, and should be revised to make the table easier to understand.

Author Response

Answering to Reviewer 2 ...

Query 1.  Although a variety of research results are presented in a comprehensive manner, we believe that the following revisions will make the paper more meaningful to readers. Especially, it would be desirable to make the tables easier for the reader to understand.

Answer: We appreciate this observation. Tables 1 and 3 were improved and we added other 2 tables for better comprehension 

Query 2.  It is necessary to be improved, as there is little mention of the method and timing of administration, which is important in the treatment of spinal cord injury.  Various cell sources useful for spinal cord injury have been clinically researched in humans. For each of the cells presented by the authors, it should be clearly stated to what extent the animal experiments and clinical trials have revealed what is known about the cells. In this point, Table 1 is not clear, which should be corrected. Table 2 also contains long sentences, which are difficult to understand for the reader, and should be revised to make the table easier to understand.

Answer:  We have added the asked information (method and timing administration) to Table 3 and improved the asked tables (1 and 2 now Table 3). Table 3 was rephrased in order to present more comprehensive sentences. However, a few sentences might remain relatively lengthy to maintain the necessary information.

Regards, 

The Authors

Round 2

Reviewer 1 Report

None